# Study of Biological Activities and ADMET-Related Properties of Salicylanilide-Based Peptidomimetics

**DOI:** 10.3390/ijms231911648

**Published:** 2022-10-01

**Authors:** Dominika Pindjakova, Eliska Pilarova, Karel Pauk, Hana Michnova, Jan Hosek, Pratibha Magar, Alois Cizek, Ales Imramovsky, Josef Jampilek

**Affiliations:** 1Department of Analytical Chemistry, Faculty of Natural Sciences, Comenius University, Ilkovicova 6, 842 15 Bratislava, Slovakia; 2Institute of Organic Chemistry and Technology, Faculty of Chemical Technology, University of Pardubice, Studentska 95, 530 09 Pardubice, Czech Republic; 3Department of Infectious Diseases and Microbiology, Faculty of Veterinary Medicine, University of Veterinary Sciences Brno, Palackeho tr. 1946/1, 612 42 Brno, Czech Republic; 4Department of Pharmacology and Toxicology, Veterinary Research Institute, Hudcova 296/70, 621 00 Brno, Czech Republic; 5Department of Chemical Biology, Faculty of Science, Palacky University Olomouc, Slechtitelu 27, 783 71 Olomouc, Czech Republic

**Keywords:** salicylamide, peptidomimetics, antimicrobial activity, cytotoxicity, lipophilicity, structure–activity relationships

## Abstract

A series of eleven benzylated intermediates and eleven target compounds derived from salicylanilide were tested against *Staphylococcus aureus* ATCC 29213 and *Enterococcus faecalis* ATCC 29212 as reference strains and against three clinical isolates of methicillin-resistant *S. aureus* (MRSA) and three isolates of vancomycin-resistant *E. faecalis*. In addition, the compounds were evaluated against *Mycobacterium tuberculosis* H37Ra and *M. smegmatis* ATCC 700084. The in vitro cytotoxicity of the compounds was assessed using the human monocytic leukemia cell line THP-1. The lipophilicity of the prepared compounds was experimentally determined and correlated with biological activity. The benzylated intermediates were found to be completely biologically inactive. Of the final eleven compounds, according to the number of amide groups in the molecule, eight are diamides, and three are triamides that were inactive. 5-Chloro-2-hydroxy-*N*-[(2*S*)- 4-(methylsulfanyl)-1-oxo-1-{[4-(trifluoromethyl)phenyl]amino}butan-2-yl]benzamide (**3e**) and 5-chloro-2-hydroxy-*N*-[(2*S*)-(4-methyl-1-oxo-1-{[4-(trifluoromethyl)phenyl]amino)pentan-2-yl)benzamide (**3f**) showed the broadest spectrum of activity against all tested species/isolates comparable to the used standards (ampicillin and isoniazid). Six diamides showed high antistaphylococcal activity with MICs ranging from 0.070 to 8.95 μM. Three diamides showed anti-enterococcal activity with MICs ranging from 4.66 to 35.8 μM, and the activities of **3f** and **3e** against *M. tuberculosis* and *M. smegmatis* were MICs of 18.7 and 35.8 μM, respectively. All the active compounds were microbicidal. It was observed that the connecting linker between the chlorsalicylic and 4-CF_3_-anilide cores must be substituted with a bulky and/or lipophilic chain such as isopropyl, isobutyl, or thiabutyl chain. Anticancer activity on THP-1 cells IC_50_ ranged from 1.4 to >10 µM and increased with increasing lipophilicity.

## 1. Introduction

Natural molecules are an inexhaustible source of inspiration for designing new bioactive entities [1,2]. Salicylic acid is one of the molecules that has made its mark among medicines [3,4]. The most famous is its acetylated derivative, Aspirin^®^, used as a nonsteroidal anti-inflammatory, antipyretic, and analgesic drug that helps prevent heart attacks, ischemic strokes, and blood clots [5]. It has made it onto the World Health Organization’s List of Essential Medicines [6]. In addition to inhibiting cyclooxygenases, acetylsalicylic acid uncouples oxidative phosphorylation in mitochondria [7,8]; inhibits the transport of protons through membranes; induces the formation of NO radicals, which has a positive effect on the stimulation of immunity [9,10]; modulates signaling via NF-κB [11,12]; activates AMP-activated protein kinase [13,14]; and acetylates cellular proteins and thus regulates their functions at the post-translational level [15,16]. The functional modification of the carboxylic acid to a carboxamide or the incorporation of a phenolic group into the carbamate resulted in a great extension of the bioeffects [17,18,19,20,21,22,23,24,25,26]. The CONH and/or OCONH moieties are peptide-bond-simulating groups and can interact with many biological targets, bringing these small molecules to the attention of medicinal chemists [27,28,29].

Thus, salicylanilides/salicylcarbamates represent a group of agents that have a wide range of activities [17,18,19,20,21,22,23,24,25,26] due to their ability to influence a wide range of targets (multitargeting [30,31]) in both prokaryotic and eukaryotic cells. Derivatives in which we have specialized for a long time mainly show good antimicrobial properties [22,23,32,33,34], but they can also be used in the treatment of cancer [35,36,37,38]. This dual (anticancer and at the same time antibacterial) effect [39,40,41,42,43] is especially advantageous in the treatment of oncology patients when treatment results in overall immunosuppression, when, in addition to nosocomial pathogens, banal infections and opportunistic pathogens also become a threat to these patients. The effects of various derivatives, especially against Gram-positive bacteria and mycobacteria, have been described earlier [22,23,32,33,34]. On the other hand, for example, the effects on the parasites *Onchocerca* or *Toxoplasma* have been described recently [44,45]. Important derivatives include, for example, niclosamide, which, in addition to its anthelmintic activity, has proven to be an important aid in the fight against cancer growth [46] as well as an antimicrobial agent [47]. The activity of all these compounds is closely related to the presence of halogen in the molecule. According to the literature, derivatives containing fluorine showed the highest biological activities, and it is likely that electronegativity is directly related to antibacterial activity (F > Cl > Br > I). In most cases, these agents are more effective against Gram-positive bacteria than against Gram-negative bacteria [32,33,48,49,50].

In one of our previous works, compounds of the “diamide” type, not only simple salicylanilides, were prepared and tested against a battery of Gram-positive bacteria and mycobacteria [33,34]. One of these agents, containing 4-(trifluoromethyl)aniline and an isopropyl chain as substituents, showed the highest activity against *Staphylococcus aureus*, methicillin-resistant *S. aureus* (MICs ranged 4.82–9.64 µM), *Bacillus cereus* (MIC 2.41 µM), *Clostridium perfringens* (MIC 4.82 µM), *Mycobacterium kansasii* (MIC 38.6 µM), and *M. smegmatis* (MIC 77.1 µM) [33]; therefore, we decided to prepare a series of derivatives from this lead compound (compound **3c** here) and subject them to deeper investigation. The investigated agents were substituted on the C_(5)_ salicyl part of the molecule with chlorine and on the anilide C_(4)_’ part with a CF_3_ moiety. The middle part was modified by various inserted amino acids, i.e., by aliphatic and aromatic substitution, and a series of new derivatives were prepared and tested for their antibacterial properties. Moreover, other extended derivatives, in which the middle fragment between the salicylic and anilide parts was prolonged by two amino acid fragments (“diamides” were changed/prolonged to “triamides”)—see Figure 1—were also tested for anti-infective activity. Thus, this contribution captures new knowledge about the anti-invasive activity of diamides and selected triamides.

## 2. Results and Discussion

### 2.1. Synthesis and Physicochemical Properties

The procedure for the preparation of acid **1** was described in a previous paper [36]. This monopeptide acid **1** was used as a starting material, which was coupled with 4-trifluoromethyl aniline to yield benzylated intermediates **2a**–**h**, whose debenzylation by hydrogen gave targeted diamides **3a**–**h** (see Figure 1). The preparation of triamides was described by Jorda et al. [38] and is briefly shown in Figure 2. The synthesis again starts from monopeptide acids **1f**, **1h**, which reacted with the ester of the second amino acid to give dipeptide acids **4a**–**c**, which coupled with 4-trifluoromethyl aniline gave benzylated triamides **5a**–**c**, which after hydrogenation debenzylation afforded the target triamides **6a**–**c**. Detailed syntheses of diamides **3f**, **3h** and triamides **7a**–**c** was published by Jorda et al. [38].

Target compounds **3a**–**h** and **6a**–**c**, including their benzyl-protected precursors **2a**–**h** and **5a**–**c**, were studied for their lipophilicity; one of the most important physicochemical parameters had a fundamental influence on the effect of bioactive molecules [51,52,53]. Experimental lipophilicity values of benzylated precursors **2** and **5** are shown in Appendix A in Appendix A, and values of final compounds **3** and **6** are shown in Table 1. In addition to the standard logarithm of the capacity factor (log *k*), lipophilicity was also expressed as the logarithm of the distribution coefficient (log *D*) at physiological pH of 6.5 and 7.4. Logically benzylated derivatives are more lipophilic than debenzylated ones (see Appendix A and Table 1); the range of averaged values of all three experimentally determined lipophilicity descriptors (log *k*/log *D*_6.5_/log *D*_7.4_) of benzylated diamides **2a**–**h** and **5a**–**c** is from 1.47 ± 0.02 (**2a**) to 2.12 ± 0.03 (**5c**). Based on the deviations, the values of log *k*, log *D*_6.5_, and log *D*_7.4_ are very close to each other, similarly to final compounds **3a**–**h** and **6a**–**c**, where the range of averaged lipophilicity values of log *k*/log *D*_6.5_/log *D*_7.4_ parameters is from 0.81 ± 0.05 (**3a**) to 1.62 ± 0.05 (**3g**). Thus, the least lipophilic compound within the discussed series is the methyl (**3a**) derivative, while cyclohexylmethyl-substituted compound **3g** is the most lipophilic derivative within a series of the final compounds, in contrast to the benzylated series, where “tribenzylated” triamide **5c** was the most lipophilic. Triamides **6a**–**c** have lipophilicity values ranging from 1.51 ± 0.05 to 1.54 ± 0.05; insignificantly increasing in the order of **6b** (R^1^ = Bn, R^2^ = iBu) < **6a** (R^1^ = iBu, R^2^ = Bn) < **6c** (R^1^ = Bn, R^2^ = Bn). It is evident from the results (Table 1) that the lipophilicity values of derivatives **3d** (R^1^ = butyl) and **3b** (R^1^ = propyl) are higher than their branched isomers **3f** (R^1^ = isobutyl) and **3c** (R^1^ = isopropyl). The lipophilicity of thiabutyl-substituted compound **3e** is approx. the same as that of the isopropyl (**3c**) derivative. The lipophilicity of the benzyl-substituted compound (**3h**) is between thiabutyl **3e** and isobutyl **3f** derivatives.

In addition to experimental values, lipophilicity values predicted by commercially available programs, such as ACD/Percepta ver. 12 (log *P*) and ChemBioDraw Ultra 13.0 (log *P*, Clog *P*), are listed in Table 1. While the mutual agreement of the experimental data has an average correlation coefficient of r = 0.996 ± 0.001 (*n* = 3 × 11), see Appendix A, the mutual correlations of the experimental and predicted values are lower (see Appendix A), indicating intermolecular and intramolecular interactions that are not captured by computational programs.

Table 1 also shows the predicted (ACD/Percepta) parameters of Lipinski’s Rule of Five (Ro5) [54], which is one of the generally accepted recommendations regarding the physicochemical parameters of biologically active agents. Ro5 contains specific molecular descriptor limits (Table 1) determined based on experimentally and statistically obtained results such that a compound meeting this recommendation is drug-like. However, it must be noted that a good drug-like score does not make a molecule a drug and vice versa [55]. Based on the data from Table 1, it can be stated that triamides significantly exceed the recommended molecular weight and lipophilicity (log *P*) values, while only some diamides do not meet the recommended log *P* values (depending on the type of software/algorithm with which log *P* is calculated). In addition, Table 1 also shows the calculated (ACD/Percepta) steric parameters (bulkiness of R^1^ and R^2^ substituents expressed as molar volumes (MV [cm^3^])) describing the length/branching of the substituents.

### 2.2. In Vitro Antimicrobial Activity

All the investigated compounds were tested for in vitro antibacterial activity against the susceptible reference strains *Staphylococcus aureus* ATCC 29213 and *Enterococcus faecalis* ATCC 29212 and representatives of multidrug-resistant clinical isolates of methicillin-resistant *S. aureus* (MRSA), SA 3202, SA 630, and 63718 carriers of the *mecA* gene [22,34], and three isolates of *vanA* gene-carrying vancomycin-resistant *E. faecalis* (VRE), 342B, 368, and 725B [56]. In addition, all the compounds were evaluated in vitro against slow-growing *Mycobacterium tuberculosis* H37Ra/ATCC 25177 and fast-growing *M. smegmatis* ATCC 700084. Activities are expressed as the minimum inhibitory concentrations (MICs) and the minimum bactericidal concentrations (MBCs); see Table 2. To establish that a compound demonstrates a bactericidal effect against a particular tested strain, it must meet the condition MIC/MBC ≤ 4 [22,34,57]. MBC values that meet this requirement, i.e., the compound is bactericidal, are indicated in bold in Table 2.

As mentioned previously, lead compound **3c** was previously evaluated for its antistaphylococcal activity [33,34], other antibacterial effects listed in Table 2 are still unpublished, and it was the inspiration for the whole series of compounds investigated here. Both benzylated precursors **2a**–**h**, **5a**–**c** and final diamides **3a**–**h** and triamides **6a**–**c** were tested in vitro for their biological activities. The blockade (benzylation) of the phenolic moiety was found to completely eliminate any effects; MICs of all the benzylated precursors were >256 µg/mL; therefore, derivatives **2** and **5** are listed in Appendix A.

Table 2 shows the antimicrobial activities of derivatives **3** and **6**. Compounds **3a**, **3b,** and **6a**–**c** were inactive compared to **3c**–**3h**. Derivatives **3e** (R^1^ = 3-thiabutyl) and **3f** (R^1^ = isobutyl) demonstrated complex antibacterial activity not only against *S. aureus*/MRSA but also against *E. faecalis*/VRE and both mycobacteria much more pronounced than the used standards. Compounds **3g** (R^1^ = cyclohexylmethyl) and **3h** (R^1^ = benzyl) showed high selective antistaphylococcal activity. It should be noted that compounds **3c**–**3h** showed comparable antistaphylococcal activities both against methicillin-susceptible *S. aureus* and MRSA isolates; therefore, it can be assumed that the presence of the *mecA* gene (which encodes an alternative transpeptidase and causes methicillin resistance [56,58]) [22,34] in MRSA does not affect the activity of these compounds. It is important to mention that all effective compounds demonstrated bactericidal activity as described for lead structure **3c** by Zadrazilova [34]. Thus, it can be speculated concerning the specific activity against *Staphylococcus* sp. Similarly, the close activity of the compounds against both *E. faecalis* and VRE indicates a mechanism of action unrelated to vancomycin resistance [56].

In an effort to clarify the mechanism of antibacterial activity, an MTT test of the most active diamides was performed. The MTT assay can be used to assess cell growth by measuring respiration. The MTT measured the viability of bacterial cells less than 70% after exposure to the MIC values for each tested compound considered a positive result of this assay. This low level of cell viability indicates the inhibition of cell growth by the inhibition of respiration [59,60]. It can be noted that compounds **3g** and **3f** showed a decrease in viability of <70% at their MIC value or below, suggesting that these agents may act through the inhibition of the respiratory chain. Compounds **3e** and **3h** inhibited the respiratory chain by 94% at a value of 2× MIC, so they are able to significantly affect it, compared to, e.g., ciprofloxacin or ampicillin. The lowest multiples of MIC values that achieved more than 70% inhibition of *S. aureus* viability (%) are shown in Table 3.

It was hypothesized that these diamides (and triamides) could behave as unnatural peptidomimetics mimicking the overall amphipathic structures and hence could have a mechanism of action similar to antimicrobial peptides [61,62]. Therefore, triamides based on the structure of active diamides were tested for their antimicrobial activity, and their zero antibacterial effect was a complete disappointment. At least the bactericidal effect of diamides was tested as their ability to disrupt bacterial membranes. Thus, an alteration in the membrane permeability of *S. aureus* ATCC 29123 was detected by a hydrophobic crystal violet dye assay [63,64]. The bacterial suspensions were treated with compounds **3e**, **3f**, **3g**, and **3h** (4× MIC) for 1 h. The uptake of crystal violet was expressed as a percentage compared to the original crystal violet solution. The results are shown in Figure 2. The test compounds did not affect plasma membrane permeability of *S. aureus* because the percentage absorption of crystal violet was comparable to the growth control and disproportionately lower than the 1% Tween 20 solution used as the positive control. Based on these results, it can be claimed that the investigated compounds do not increase membrane permeability.

### 2.3. In Vitro Cell Viability

Preliminary in vitro cytotoxicity screening of all the investigated compounds, **2**, **3**, **5**, and **6,** was performed using human monocytic leukemia cell line THP-1 in the culture medium containing 10% FBS, and it was expressed as IC_50_ values (see Table 2). Similar to the case of antimicrobial activity, no activity was observed for any of the benzylated derivatives (IC_50_ > 10 µM, see Appendix A). These findings are consistent with previously published results for benzylated precursors [38]. It should be mentioned that compounds **3a**, **3f**, and **3h** and all triamides **6a**–**c** were investigated for their anticancer potential [35,38], where, for example, their IC_50_ on the chronic myeloid leukemia cell line K562 or the human breast carcinoma cell MCF-7 ranged from approx. 3 to 6 μM (see [35,38] for detailed results).

Antimicrobially active diamides **3d**, **3f**–**h** showed IC_50_ values from 1.4 to 4.5 µM on THP-1 cells. Any cytotoxic effect (IC_50_ > 10 µM) was found for **3e** (R^1^ = thiabutyl) active in the entire spectrum of evaluated bacteria and for previously studied **3c** (R^1^ = isopropyl). However, based on all these observations, it can be concluded that high antibacterial activity is associated with a significant antiproliferative effect against cancer cell lines, whereas, according to previous studies, diamides are able to inhibit DNA replication without any effect on protein expression; activate apoptosis as well as autophagy; and can modulate the attachment of cancer cells and FAK signaling, which is important for cell survival, proliferation, and migration [35,38]. These facts make the discussed diamides interesting anti-invasive agents with dual antiproliferative and antibacterial activity.

### 2.4. Structure–Activity Relationships

As mentioned previously, biologically active diamides show rather higher log *P* (depending on the type of software) compared to what is recommended in Ro5, which, however, seems to be important for their antibacterial activity and also affects the antiproliferative effect. In previous studies of diamides and their antibacterial activity, it was clearly demonstrated that for significant efficacy, the middle part of the molecule must be substituted by a branched chain, i.e., isopropyl or benzyl. Compounds with no substitution (R^1^ = H) or substituted with methyl, propyl, or nitrogen heterocycle resulted in a significant reduction in antimicrobial activity [32,33]. Based on the results listed in Table 2, the isopropyl chain (compound **3c**, log *k* = 1.081, MV = 64.18 cm^3^) also proved to be effective against *E. feacalis*/VRE and mycobacteria, although better results were obtained with the slightly less lipophilic but bulkier thiabutyl (compound **3e**, log *k* = 1.076, MV = 77.77 cm^3^), and especially the bulkier and more lipophilic isobutyl chain (compound **3f**, log *k* = 1.266, MV = 80.68 cm^3^). Compounds **3g** and **3h** substituted with a cyclohexylmethyl (log *k* = 1.679, MV = 113.26 cm^3^) or benzyl (log *k* = 1.192, MV = 91.49 cm^3^) ring exceeded the optimal lipo-hydrophilic properties; on the other hand, the butyl chain (compound **3d**, log *k* = 1.322, MV = 80.31 cm^3^) did not reach the optimum and were only effective against *Staphylococcus* sp. The optimal lipophilicity and bulkiness parameters of the substituents for antibacterial activity appear to be approx. MV ~80 cm^3^ and log *k* ~1.1. Finally, it is important to note that the antiproliferative effect against cancer cells increases with increasing lipophilicity, where derivatives **3c** and **3e** with log *k* 1.081 and 1.076, respectively, have IC_50_ >10 µM, while compound **3h** with log *k* 1.192 has IC_50_ 3.3 µM.

## 3. Materials and Methods

### 3.1. General Methods

All reagents and solvents were purchased from commercial sources: Sigma-Aldrich/Merck (Prague, Czech Republic), Lach-Ner (Neratovice, Czech Republic), Fluorochem (Hadfield, UK), TCI Europe (Zwijndrecht, Belgium), Acros Organics (Geel, Belgium). Commercial-grade reagents were used without further purification. Reactions were monitored by using thin-layer chromatography (TLC) plates coated with 0.2 mm silica gel (60 F254, Merck, Darmstadt, Germany). TLC plates were visualized using UV irradiation (254 nm). All melting points were determined by using a Melting Point B-540 apparatus (Büchi, Flawil, Switzerland) and are given in their uncorrected form. The IR spectra were recorded with a Nicolet 6700 FTIR spectrometer (Thermo Fisher Scientific, Waltham, MA, USA) over the range 4000–400 cm^1^ by using the ATR technique. The NMR spectra were measured in DMSO-*d*_5_ or CDCl_3_-*d*_3_ solutions at ambient temperature with a Bruker Avance™ III 400 spectrometer (Bruker, Ettlingen, Germany) at frequencies 400 MHz (1H) and 100.26 MHz (13C), or with a Bruker Ascend™ 500 spectrometer at frequencies 500.13 MHz (1H) and 125.76 MHz (13C{1H}). The chemical shifts are given in ppm and are related to the following residual solvent peaks: ~2.49 (DMSO-*d*_5_), ~7.27 (CDCl_3_). The coupling constants (*J*) are reported in Hz. Elemental (CHN) analyses were performed with an automatic microanalyzer Flash 2000 (Thermo Scientific, West Palm Beach, FL, USA). Mass spectrometry with high resolution was determined by the “dried droplet” method using a MALDI mass spectrometer LTQ Orbitrap XL (Thermo Scientific) equipped with a nitrogen UV laser (337 nm, 60 Hz). Spectra were measured in positive ion mode and in regular mass extent with a resolution of 100,000 at a mass-to-charge ratio (*m*/*z*) of 400, with 2,5-dihydrobenzoic acid (DBH) used as the matrix.

### 3.2. Synthesis

#### 3.2.1. General Procedure for Synthesis of (benzyloxy) Trifluoromethylbenzamides **2**

Carboxylic acid **1** (1 mM) was dissolved in dichloromethane (DCM, 30 mL) and hydroxybenzotriazole (HOBt, 1.1 mM), and 1-ethyl-3-(3-dimethylaminopropyl)carbodiimide hydrochloride (EDC∙HCl) (0.95 mM) was added in one portion. Reaction mixture was stirred for 1 h, and then 4-(trifluoromethyl)aniline (1 mM) in DCM (10 mL) was added. Reaction mixture was stirred for 18 h at RT. DCM was removed by rotary evaporation, and residue was dissolved in ethylacetate (EtOAc, 30 mL). Solution was extracted by saturated solution of NaHCO_3_ (3 × 20 mL), 5% citric acid (3 × 20 mL), and saturated NaCl (20 mL). (Benzyloxy)trifluoromethylbenzamide (**2**) was purified via crystallization and/or column chromatography.

*2-(B**enzyloxy)-5-chloro-N-[(2S)-1-oxo-1-{[4-(trifluoromethyl)phenyl]amino}propan-2-yl]benzamide* (**2a**). Reaction was completed on a 3.6 mM scale. White crystals; yield 75%; mp = 140.9–143.2 °C; R_f_ = 0.633 (EtOAc/n-hexane 1:1). ^1^H NMR (500 MHz, DMSO): δ 10.48 (s, 1H, N*H*), 8.58 (d, J = 6.7 Hz, 1H, N*H*), 7.82 (d, J = 8.4 Hz, 2H, 2×C*H*(Ar)), 7.76 (d, J = 2.0 Hz, 1H, C*H*(Ar)), 7.69 (d, J = 8.5 Hz, 2H, 2×C*H*(Ar)), 7.56 (d, J = 7.8 Hz, 3H, 3×C*H*(Ar)), 7.37 (dt, J = 24.6, 8.1 Hz, 4H, 4×C*H*(Ar)), 5.33–5.26 (m, 2H, O-C*H*_2_-Ph), 4.62 (p, J = 6.9 Hz, 1H, NH-C*H*-C=O), 1.28 (d, J = 7.0 Hz, 3H, C*H*_3_). ^13^C NMR (100 MHz, DMSO): δ 172.15, 163.91, 155.69, 143.09, 136.54, 132.63, 130.48, 129.15, 128.90, 128.68, 126.73, 126.70, 125.45, 124.90, 124.09 (q, ^1^J (^19^F,^13^C) = 32 Hz), 119.80, 116.39, 71.47, 50.38, 18.65. ^19^F NMR (400 MHz, CDCl_3_): δ -60.34 (s). CHN analysis: Calc. For C_24_H_20_ClF_3_N_2_O_3_ (476.88): C, 60.45; H, 4.23; N, 5.87. Found: C, 60.08 ± 0.22; H, 4.15 ± 0.12; N, 5.48 ± 0.02. HRMS: m/z calc. for C_24_H_20_ClF_3_N_2_O_3_: 477.11928 [M+H]^+^, 499.10122 [M+Na]^+^, 515.07516 [M+K]^+^; found: 477.11873 [M+H]^+^, 499.10068 [M+Na]^+^, 515.07461 [M+K]^+^.

*2-(B**enzyloxy)-5-chloro-N-[(2S)-1-oxo-1-{[4-(trifluoromethyl)phenyl]amino}pentan-2-yl]benzamide* (**2b**). Reaction was completed on a 4.5 mM scale. White crystals; yield 59%; mp = 160.3–161.6 °C; R_f_ = 0.636 (EtOAc/n-hexane 1:1). ^1^H NMR (400 MHz, DMSO): δ 10.50 (s, 1H, N*H*), 8.44 (d, J = 7.3 Hz, 1H, N*H*), 7.81 (d, J = 8.5 Hz, 2H, 2×C*H*(Ar)), 7.75 (d, J = 2.0 Hz, 1H, C*H*(Ar)), 7.67 (d, J = 8.4 Hz, 2H, 2×C*H*(Ar)), 7.59–7.51 (m, 3H, 3×C*H*(Ar)), 7.43–7.32 (m, 4H, 4×C*H*(Ar)), 5.31–5.23 (m, 2H, O-C*H*_2_-Ph), 4.59 (dd, J = 13.1, 7.6 Hz, 1H, NH-C*H*-C=O), 1.63 (dt, J = 14.7, 5.9 Hz, 1H,C*H*H), 1.53–1.42 (m, 1H, CH*H*), 1.24–1.08 (m, 2H, C*H*_2_), 0.76 (t, J = 7.3 Hz, 3H, C*H*_3_).^13^C NMR (100 MHz, CDCl_3_): δ 171.67, 164.08, 155.77, 143.04, 136.37, 132.65, 130.53, 129.17, 129.00, 128.96, 126.70, 126.67, 125.47, 124.78, 124.10 (q, ^1^J (^19^F,^13^C) = 31 Hz), 119.85, 116.30, 71.63, 54.51, 34.67, 19.07, 14.22. ^19^F NMR (400 MHz, CDCl_3_): δ -60.36 (s). CHN analysis: Calc. For C_26_H_24_ClF_3_N_2_O_3_ (504.93): C, 61.85; H, 4.79; N, 5.55. Found: C, 61.51 ± 0.29; H, 4.81 ± 0.02; N, 5.54 ± 0.02. HRMS: m/z calc. for C_26_H_24_ClF_3_N_2_O_3_: 505.15003 [M+H]^+^, 527.13198 [M+Na]^+^; found: 505.15155 [M+H]^+^, 527.13354 [M+Na]^+^.

*2-(B**enzyloxy)-5-chloro-N-[(2S)-3-methyl-1-oxo-1-{[4-(trifluoromethyl)phenyl]amino}butan-2-yl]benzamide* (**2c**). Reaction was completed on a 4.8 mM scale. White crystals; yield 67%; mp = 195.2–196.5 °C; R_f_ = 0.536 (EtOAc/n-hexane 1:1). ^1^H NMR (500 MHz, CDCl_3_): δ 9.51 (s, 1H, N*H*), 8.54 (d, J = 7.7 Hz, 1H, N*H*), 8.15 (d, J = 2.8 Hz, 1H, C*H*(Ar)), 7.57 (d, J = 8.5 Hz, 2H, 2×C*H*(Ar)), 7.50 (dd, J = 7.8, 1.4 Hz, 2H, 2×C*H*(Ar)), 7.46–7.39 (m, 6H, 6×C*H*(Ar)), 7.07 (d, J = 8.9 Hz, 1H, C*H*(Ar)), 5.21 (dd, J = 28.5, 10.5 Hz, 2H, O-C*H*_2_-Ph), 4.63 (t, J = 7.4 Hz, 1H, NH-C*H*-C=O), 2.05–1.96 (m, 1H, C*H*-(CH_3_)_2_), 0.90 (d, J = 6.7 Hz, 3H, C*H*_3_), 0.66 (d, J = 6.8 Hz, 3H, C*H*_3_). ^13^C NMR (100 MHz, CDCl_3_): δ 170.03, 164.80, 155.71, 141.13, 134.49, 133.13, 131.94, 129.25, 129.13, 128.73, 126.93, 125.97, 125.54 (q, ^1^J (^19^F,^13^C) = 24 Hz),125.4, 122.05, 119.26, 114.13, 72.05, 60.46, 30.34, 19.35, 18.01. ^19^F NMR (400 MHz, CDCl_3_): δ -62.15 (s). CHN analysis: Calc. For C_26_H_24_ClF_3_N_2_O_3_ (504.93): C, 61.85; H, 4.79; N, 5.55. Found: C, 62.23 ± 0.05; H, 4.75 ± 0.03; N, 5.42 ± 0.04. HRMS: m/z calc. for C_26_H_24_ClF_3_N_2_O_3_: 505.15058 [M+H]^+^, 527.13252 [M+Na]^+^, 543.10646 [M+K]^+^; found: 505.15036 [M+H]^+^, 527.13226 [M+Na]^+^, 543.10602 [M+K]^+^.

*2-(B**enzyloxy)-5-chloro-N-[(2S)-1-oxo-1-{[4-(trifluoromethyl)phenyl]amino}hex**an-2-yl]benzamide* (**2d**). Reaction was completed on a 17.1 mM scale. White crystals; yield 4%; mp = 141.4–142.4 °C; R_f_ = 0.622 (EtOAc/n-hexane 1:1). ^1^H NMR (500 MHz, DMSO): δ 10.53 (s, 1H, N*H*), 8.49 (d, J = 7.3 Hz, 1H, N*H*), 7.82 (d, J = 8.4 Hz, 2H, 2×C*H*(Ar)), 7.76 (d, J = 2.7 Hz, 1H, C*H*(Ar)), 7.68 (d, J = 8.5 Hz, 2H, 2×C*H*(Ar)), 7.57 (dd, J = 11.4, 5.0 Hz, 3H, 3×C*H*(Ar)), 7.42 – 7.32 (m, 4H, 4×C*H*(Ar)), 5.32–5.24 (m, 2H, O-C*H*_2_-Ph), 4.58 (dd, J = 13.2, 7.6 Hz, 1H, NH-C*H*-C=O), 1.71–1.62 (m, 1H, C*H*H), 1.54–1.43 (m, 1H, CH*H*), 1.24–1.06 (m, 4H, C*H*_2_-C*H*_2_), 0.75 (t, J = 6.6 Hz, 3H, C*H*_3_). ^13^C NMR (100 MHz, DMSO): δ 171.49, 163.87, 155.55, 142.84, 136.21, 132.51, 130.34, 128.98, 128.82, 128.71, 126.57, 125.26, 124.80 (q, ^1^J (^19^F,^13^C) = 74 Hz), 124.01, 123.75, 119.62, 116.08, 71.34, 54.52, 32.14, 27.76, 22.26, 14.21. ^19^F NMR (100 MHz, DMSO): δ -60.37 (s). CHN analysis: Calc. For C_27_H_26_ClF_3_N_2_O_3_ (518.96): C, 62.49; H, 5.05; N, 5.40. Found: C, 62.38 ± 0.06; H, 4.92 ± 0.05; N, 5.03 ± 0.03. HRMS: m/z calc. for C_27_H_26_ClF_3_N_2_O_3_: 541.14763 [M+Na]^+^; found: 541.14899 [M+Na]^+^.

*2-(B**enzyloxy)-5-chloro-N-[(2S)-4-(methylsulfanyl)-1-oxo-1-{[4-(trifluoromethyl)phenyl]amino}butan-2-yl]benzamide* (**2e**). Reaction was completed on a 7.3 mM scale. White crystals; yield 23%; mp = 171.6–173.2 °C; R_f_ = 0.688 (EtOAc/n-hexane 1:1). ^1^H NMR (500 MHz, DMSO): δ 10.56 (s, 1H, N*H*), 8.58 (d, J = 7.5 Hz, 1H, N*H*), 7.82 (d, J = 8.5 Hz, 2H, 2×C*H*(Ar)), 7.73–7.66 (m, 3H, 3×C*H*(Ar)), 7.59–7.51 (m, 3H, 3×C*H*(Ar)), 7.36 (dt, J = 16.7, 8.0 Hz, 4H, 4×C*H*(Ar)), 5.31–5.23 (m, 2H, O-C*H*_2_-Ph), 4.74–4.66 (m, 1H, NH-C*H*-C=O), 2.42–2.27 (m, 2H, C*H*_2_), 2.03–1.93 (m, 4H, C*H*_3_, C*H*H), 1.87–1.77 (m, 1H, CH*H*). ^13^C NMR (100 MHz, DMSO): δ 170.93, 164.32, 155.42, 142.79, 136.35, 132.34, 130.17, 129.00, 128.77, 128.53, 126.56, 125.15, 125.03, 123.96 (q, ^1^J (^19^F,^13^C) = 25 Hz), 123.84, 119.78, 116.05, 71.18, 53.81, 31.99, 29.84, 15.06. ^19^F NMR (400 MHz, DMSO): δ -60.35 (s). CHN analysis: Calc. For C_26_H_24_ClF_3_N_2_O_3_S (536.99): C, 58.15; H, 4.50; N, 5.22. Found: C, 58.62 ± 0.15; H, 4.40 ± 0.02; N, 5.20 ± 0.06. HRMS: m/z calc. for C_26_H_24_ClF_3_N_2_O_3_S: 559.10405 [M+Na]^+^; found: 559.10553 [M+Na]^+^.

*2-(B**enzyloxy)-5-chloro-N-[(2S)-3-cyclohexyl-1-oxo-1-{[4-(trifluoromethyl)phenyl]amino}propan-2-yl]benzamide* (**2g**). Reaction was completed on a 12.3 mM scale. White crystals; yield 56%; mp = 166.4–167.4 °C; R_f_ = 0.663 (EtOAc/n-hexane 1:1). ^1^H NMR (500 MHz, DMSO): δ 10.53 (s, 1H, N*H*), 8.44 (d, J = 7.5 Hz, 1H, N*H*), 7.82 (d, J = 8.6 Hz, 2H, 2×C*H*(Ar)), 7.75 (d, J = 2.8 Hz, 1H, C*H*(Ar)), 7.68 (d, J = 8.7 Hz, 2H, 2×C*H*(Ar)), 7.59–7.53 (m, 3H, 3×C*H*(Ar)), 7.42–7.32 (m, 4H, 4×C*H*(Ar)), 5.32–5.25 (m, 2H, O-C*H*_2_-Ph), 4.65 (ddd, J = 9.8, 7.6, 5.2 Hz, 1H, NH-C*H*-C=O), 1.68 (d, J = 12.3 Hz, 1H, C*H*-CH_2_), 1.63–1.45 (m, 5H (*cyclohexylalanine*)), 1.41–1.33 (m, 1H (*cyclohexylalanine*)), 1.23–1.12 (m, 1H (*cyclohexylalanine*)), 1.09–0.97 (m, 3H (*cycylohexylalanine*)), 0.88–0.73 (m, 2H, C*H*_2_-CH). ^13^C NMR (100 MHz, DMSO): δ 172.02, 163.89, 155.55, 142.90, 136.27, 132.52, 130.32, 129.03, 128.84, 128.61, 126.55, 125.26, 124.81 (q, ^1^J (^19^F,^13^C) = 67 Hz), 123.74, 119.67, 116.09, 71.30, 52.40, 34.10, 33.40, 32.17, 26.36, 26.10, 25.95. ^19^F NMR (400 MHz, DMSO): δ -60.36 (s). CHN analysis: Calc. For C_30_H_30_ClF_3_N_2_O_3_ (559.02): C, 64.46; H, 5.41; N, 5.01. Found: C, 64.53 ± 0.06; H, 5.36 ± 0.03; N, 5.24 ± 0.02. HRMS: m/z calc. for C_30_H_30_ClF_3_N_2_O_3_: 581.17893 [M+Na]^+^; found: 581.18066 [M+Na]^+^.

#### 3.2.2. General Procedure for Synthesis of (hydroxy) Trifluoromethylbenzamides 3

Benzyl-protected trifluoromethylbenzamide **2** (1 mM) was dissolved in EtOAc (50 mL), and palladium with 10% on carbon was added (catalytic amount). The reaction mixture was stirred overnight in a hydrogen atmosphere to complete conversion of starting materials (TLC control). Catalyst was removed, and ethyl acetate was evaporated under reduced pressure. Obtained residue was purified by column chromatography on silica gel (n-hexane/EtOAc 5:1) to give compound **3**.

*5-C**hloro-2-hydroxy-N-[(2S)-1-oxo-1-{[4-(trifluoromethyl)phenyl]amino}propan-2-yl]benzamide* (**3a**). Reaction was completed on a 1.8 mM scale. White crystals; yield 29%; mp = 240.3–241.2 °C; R_f_ = 0.505 (EtOAc/n-hexane 1:3, evaluated 3×). ^1^H NMR (400 MHz, DMSO): δ 12.20 (s, 1H, O*H*), 10.52 (s, 1H, N*H*), 9.12 (d, J = 6.5 Hz, 1H, N*H*), 8.07 (d, J = 2.2 Hz, 1H, C*H*(Ar)), 7.84 (d, J = 8.4 Hz, 2H, 2×C*H*(Ar)), 7.69 (d, J = 8.5 Hz, 2H, 2×C*H*(Ar)), 7.45 (dd, J = 8.8, 2.3 Hz, 1H, C*H*(Ar)), 6.98 (d, J = 8.8 Hz, 1H, C*H*(Ar)), 4.68 (p, J = 6.8 Hz, 1H, NH-C*H*-C=O), 1.48 (d, J = 7.0 Hz, 3H, C*H*_3_). ^13^C NMR (100 MHz, DMSO): δ 172.09, 167.09, 158.43, 143.09, 133.95, 128.93, 126.74, 124.32, 124.13 (q, ^1^J (^19^F,^13^C) = 32 Hz), 123.25, 119.90, 119.83, 117.94, 50.50, 18.57. ^19^F NMR (400 MHz, DMSO): δ -60.38 (s). CHN analysis: Calc. For C_17_H_14_ClF_3_N_2_O_3_ (386.75): C, 52.79; H, 3.65; N, 7.24. Found: C, 52.92 ± 0.06; H, 3.67 ± 0.03; N, 7.10 ± 0.03. HRMS: m/z calc. for C_17_H_14_ClF_3_N_2_O_3_: 387.07178 [M+H]^+^, 409.05373 [M+Na]^+^; found: 387.07315 [M+H]^+^, 409.05508 [M+Na]^+^.

*5-C**hloro-2-hydroxy-N-[(2S)-1-oxo-1-{[4-(trifluoromethyl)phenyl]amino}pentan-2-yl]benzamide* (**3b**). Reaction was completed on a 0.7 mM scale. White crystals; yield 28%; mp = 216.5–218.3 °C; R_f_ = 0.533 (EtOAc/n-hexane 1:3, evaluated 3×). ^1^H NMR (400 MHz, DMSO): δ 12.18 (s, 1H, O*H*), 10.56 (s, 1H, N*H*), 9.03 (d, J = 7.0 Hz, 1H, N*H*), 8.06 (d, J = 2.6 Hz, 1H, C*H*(Ar)), 7.83 (d, J = 8.4 Hz, 2H, 2×C*H*(Ar)), 7.67 (d, J = 8.4 Hz, 2H, 2×C*H*(Ar)), 7.44 (dd, J = 8.8, 2.6 Hz, 1H, C*H*(Ar)), 7.05–6.82 (m, 1H, C*H*(Ar)), 4.79–4.47 (m, 1H, NH-C*H*-C=O), 2.00–1.71 (m, 2H, C*H*_2_), 1.58–1.28 (m, 2H, C*H*_2_), 1.00–0.85 (m, 3H, C*H*_3_). ^13^C NMR (100 MHz, DMSO): δ 171.66, 167.19, 158.26, 142.99, 133.90, 128.99, 126.73, 126.70, 124.16 (q, ^1^J (^19^F,^13^C) = 32 Hz), 123.28, 119.89, 119.80, 118.08, 54.59, 34.42, 19.43, 14.23. ^19^F NMR (400 MHz, DMSO): δ -60.40 (s). CHN analysis: Calc. For C_19_H_18_ClF_3_N_2_O_3_ (414.81): C, 55.01; H, 4.37; N, 6.75. Found: C, 55.20 ± 0.02; H, 4.49 ± 0.02; N, 6.68 ± 0.02. HRMS: m/z calc. for C_19_H_18_ClF_3_N_2_O_3_: 415.10308 [M+H]^+^, 437.08503 [M+Na]^+^; found: 415.10428 [M+H]^+^, 437.08624 [M+Na]^+^.

*5-C**hloro-2-hydroxy-N-[(2S)-3-methyl-1-oxo-1-{[4-(trifluoromethyl)phenyl]amino}butan-2-yl]benzamide* (**3c**). Reaction was completed on a 1.5 mM scale. White crystals; yield 61%; mp = 192.6–194.6 °C; R_f_ = 0.716 (EtOAc/n-hexane 1:3, evaluated 3×). ^1^H NMR (500 MHz, DMSO): δ 12.02 (s, 1H, O*H*), 10.61 (s, 1H, N*H*), 8.97 (d, J = 8.1 Hz, 1H, N*H*), 8.02 (d, J = 2.7 Hz, 1H, C*H*(Ar)), 7.84 (d, J = 8.5 Hz, 2H, 2×C*H*(Ar)), 7.69 (d, J = 8.6 Hz, 2H, 2×C*H*(Ar)), 7.44 (dd, J = 8.8, 2.7 Hz, 1H, C*H*(Ar)), 6.99 (d, J = 8.8 Hz, 1H, C*H*(Ar)), 4.59 (t, J = 7.5 Hz, 1H, NH-C*H*-C=O), 2.21 (dd, J = 13.6, 6.8 Hz, 1H, (CH3)2-C*H*), 0.98 (dd, J = 6.7, 3.1 Hz, 6H, (C*H*_3_)_2_-CH) (Appendix A). ^13^C NMR (100 MHz, CDCl_3_): δ 170.86, 165.99, 157.15, 142.61, 133.49, 129.40, 126.60, 126.57, 124.00 (q, ^1^J (^19^F,^13^C) = 26 Hz), 123.29, 119.68, 119.49, 118.84, 59.60, 31.14, 19.64, 18.82 (Appendix A). ^19^F NMR (400 MHz, DMSO): δ -60.36 (s) (Appendix A). CHN analysis: Calc. For C_19_H_18_ClF_3_N_2_O_3_ (414.81): C, 55.01; H, 4.37; N, 6.75. Found: C, 55.81 ± 0.06; H, 4.86 ± 0.06; N, 6.32 ± 0.02. HRMS: m/z calc. for C_19_H_18_ClF_3_N_2_O_3_: 437.08503 [M+Na]^+^; found: 437.08582 [M+Na]^+^.

*5-C**hloro-2-hydroxy-N-[(2S)-1-oxo-1-{[4-(trifluoromethyl)phenyl]amino}hex**an-2-yl]benzamide* (**3d**). Reaction was completed on a 1.2 mM scale. White crystals; yield 40%; mp = 201.9–203.8 °C; R_f_ = 0.496 (EtOAc/n-hexane 1:5, evaluated 2×). ^1^H NMR (500 MHz, DMSO): δ 12.20 (s, 1H, O*H*), 10.59 (s, 1H, N*H*), 9.05 (d, J = 7.2 Hz, 1H, N*H*), 8.06 (d, J = 2.5 Hz, 1H, C*H*(Ar)), 7.83 (d, J = 8.5 Hz, 2H, 2×C*H*(Ar)), 7.68 (d, J = 8.5 Hz, 2H, 2×C*H*(Ar)), 7.44 (dd, J = 8.8, 2.5 Hz, 1H, C*H*(Ar)), 6.97 (d, J = 8.8 Hz, 1H, C*H*(Ar)), 4.65 (q, J = 7.0 Hz, 1H, NH-C*H*-C=O), 1.88–1.79 (m, 2H, C*H*_2_-CH), 1.43-1.30 (m, 4H, C*H*_2_-C*H*_2_), 0.86 (t, J = 6.8 Hz, 3H, C*H*_3_) (Appendix A). ^13^C NMR (100 MHz, DMSO): δ 171.48, 166.92, 158.02, 142.80, 133.74, 128.83, 126.57, 126.55, 123.96 (q, ^1^J (^19^F,^13^C) = 25 Hz), 123.11, 119.67, 119.62, 117.89, 54.60, 31.93, 28.11, 22.35, 14.30 (Appendix A). ^19^F NMR (100 MHz, DMSO): δ -60.40 (s) (Appendix A). CHN analysis: Calc. For C_20_H_20_ClF_3_N_2_O_3_ (428.83): C, 56.02; H, 4.70; N, 6.53. Found: C, 56.29 ± 0.01; H, 4.76 ± 0.01; N, 6.28 ± 0.03. HRMS: m/z calc. for C_20_H_20_ClF_3_N_2_O_3_: 451.10068 [M+Na]^+^; found: 451.10102 [M+Na]^+^.

*5-C**hloro-2-hydroxy-N-[(2S)-4-(methylsulfanyl)-1-oxo-1-{[4-(trifluoromethyl)phenyl]amino}butan-2-yl]benzamide* (**3e**). Reaction was completed on a 1.4 mM scale. White crystals; yield 51%; mp = 153.9–155.2 °C; R_f_ = 0.588 (EtOAc/n-hexane 1:1). ^1^H NMR (500 MHz, DMSO): δ 12.21 (s, 1H, O*H*), 10.61 (s, 1H, N*H*), 9.11 (d, J = 7.2 Hz, 1H, N*H*), 8.06 (d, J = 2.5 Hz, 1H, C*H*(Ar)), 7.84 (d, J = 8.5 Hz, 2H, 2×C*H*(Ar)), 7.69 (d, J = 8.6 Hz, 2H, 2×C*H*(Ar)), 7.46 (dd, J = 8.8, 2.6 Hz, 1H, C*H*(Ar)), 6.97 (d, J = 8.8 Hz, 1H, C*H*(Ar)), 4.79–4.70 (m, 1H, NH-C*H*-C=O), 2.66–2.51 (m, 2H, C*H*_2_), 2.18–2.03 (m, 5H, C*H*_3_-S, C*H*_2_-S) (Appendix A). ^13^C NMR (100 MHz, CDCl_3_):170.88, 167.47, 158.29, 142.74, 133.86, 128.66, 126.56, 126.53, 124.02 (q, ^1^J (^19^F,^13^C) = 26 Hz), 123.02, 119.82, 119.66, 117.67, 53.96, 31.70, 30.19, 15.12 (Appendix A). ^19^F NMR (400 MHz, DMSO): δ -60.38 (s) (Appendix A). CHN analysis: Calc. For C_19_H_18_ClF_3_N_2_O_3_S (446.87): C, 51.07; H, 4.06; N, 6.27. Found: C, 50.93 ± 0.34; H, 4.48 ± 0.01; N, 5.64 ± 0.18. HRMS: m/z calc. for C_19_H_18_ClF_3_N_2_O_3_S: 469.05710 [M+Na]^+^; found: 469.05844 [M+Na]^+^.

*5-C**hloro-N-[(2S)-3-cyclohexyl-1-oxo-1-{[4-(trifluoromethyl)phenyl]amino}propan-2-yl]-2-hydroxybenzamide* (**3g**). Reaction was completed on a 1.8 mM scale. White crystals; yield 39%; mp = 217.3–218.1 °C; R_f_ = 0.474 (EtOAc/n-hexane 1:5, evaluated 2×). ^1^H NMR (500 MHz, DMSO): δ 12.25 (s, 1H, O*H*), 10.58 (s, 1H, N*H*), 9.02 (d, J = 7.5 Hz, 1H, N*H*), 8.07 (d, J = 2.5 Hz, 1H, C*H*(Ar)), 7.83 (d, J = 8.5 Hz, 2H, 2×C*H*(Ar)), 7.60 (d, J = 9.3 Hz, 2H, 2×C*H*(Ar)), 7.45 (dd, J = 8.8, 2.6 Hz, 1H, C*H*(Ar)), 6.97 (d, J = 8.8 Hz, 1H, C*H*(Ar)), 4.75–4.71 (m, 1H, NH-C*H*-C=O), 1.79-1.71 (m, 3H, 3×*H*-*cyclohexylalanine*), 1.68–1.62 (3H, m, 3×*H*-*cyclohexylalanine*), 1.59 (s, 1H, *H*-*cyclohexylalanine*), 1.40–1.39 (m, 1H, C*H*-cyclohexylalanine), 1.24–1.11 (m, 3H, 3×*H*-*cyclohexylalanine*), 0.99-0.92 (m, 2H, C*H*_2_-CH-cyclohexylalanine) (Appendix A). ^13^C NMR (100 MHz, DMSO): δ 171.96, 167.20, 158.23, 142.86, 133.84, 128.69, 126.58, 126.55, 123.93 (q, ^1^J (^19^F,^13^C) = 25 Hz), 123.08, 119.73, 119.68, 117.68, 52.44, 34.25, 32.22, 26.45, 26.21, 26.01 (Appendix A). ^19^F NMR (100 MHz, DMSO): δ -60.38 (s) (Appendix A). CHN analysis: Calc. For C_23_H_24_ClF_3_N_2_O_3_ (468.90): C, 58.91; H, 5.16; N, 5.97. Found: C, 59.46 ± 0.18; H, 5.09 ± 0.04; N, 5.72 ± 0.02. HRMS: m/z calc. for C_23_H_24_ClF_3_N_2_O_3_: 491.13198 [M+Na]^+^; found: 491.13200 [M+Na]^+^.

The synthesis and detailed characterization of compounds **2f**/**3f**, **2h**/**3h** were reported by Jorda et al. [38]. The synthesis and characterization of benzylated triamide precursors **5** and final triamides **6** were reported by Jorda et al. [38].

### 3.3. Lipophilicity Determination by HPLC

An HPLC separation module Agilent 1200 Series (Agilent Technologies, Santa Clara, CA, USA) equipped with a Dual Absorbance Detector (DAD SL G1315C, Agilent Technologies) was used. A chromatographic column Symmetry^®^ C18 5 μm, 4.6 × 250 mm, Part No. W21751W016 (Waters Corp, Milford, MA, USA) was used. The HPLC separation process was monitored by ChemStation for LC 3D systems (Agilent Technologies). Isocratic elution by a mixture of MeOH (HPLC grade, 72%) and H_2_O-HPLC Mili-Q grade (28%) as a mobile phase was used for the determination of capacity factor *k*. Isocratic elution by a mixture of MeOH (HPLC grade, 72%) and acetate-buffered saline (pH 7.4 and pH 6.5) (28%) as a mobile phase was used for the determination of distribution coefficient expressed as *D*_7.4_ and *D*_6.5_. The total flow of the column was 1.0 mL/min, injection was 20 μL, column temperature was 40 °C, and sample temperature was 10 °C. The detection wavelength of 210 nm was chosen. A KI methanolic solution was used for the determination of the dead times (*t_d_*). Retention times (*t_r_*) were measured in minutes. The capacity factors *k* were calculated according to the formula *k* = (*t_r_* − *t_d_*)/*t_d_*, where *t_R_* is the retention time of the solute, and *t_d_* is the dead time obtained using an unretained analyte. The distribution coefficients *D*_pH_ were calculated according to the formula *D*_pH_ = (*t_r_* − *t_d_*)/*t_d_*. Each experiment was repeated three times. The log *k* values of individual compounds are shown in Table 1.

### 3.4. In Vitro Antibacterial Evaluation

In vitro antibacterial activity of the synthesized compounds was evaluated against representatives of multidrug-resistant bacteria, three clinical isolates of methicillin-resistant *S. aureus*: clinical isolate of animal origin, MRSA 63718 (Department of Infectious Diseases and Microbiology, Faculty of Veterinary Medicine, University of Veterinary Sciences Brno, Czech Republic), carrying the *mecA* gene [58]; and MRSA SA 630 and MRSA SA 3202 [22] (National Institute of Public Health, Prague, Czech Republic), both of human origin. These three clinical isolates were classified as vancomycin-susceptible (but with higher MIC of vancomycin equal to 2 μg/mL (VA2-MRSA) within the susceptible range for MRSA 63718) methicillin-resistant *S. aureus* (VS-MRSA) [22]. Vancomycin- and methicillin-susceptible *S. aureus* ATCC 29213 and vancomycin-susceptible *Enterococcus faecalis* ATCC 29212, obtained from the American Type Culture Collection, were used as the reference and quality control strains. Three *vanA* gene-carrying vancomycin-resistant isolates of *E. faecalis* (VRE 342B, VRE 368, VRE 725B) were provided by Oravcova et al. [56].

The minimum inhibitory concentrations (MICs) were evaluated by the microtitration broth method according to the CLSI [65,66] with some modifications. The compounds were dissolved in DMSO (Sigma, St. Louis, MO, USA) to obtain a concentration of 10 µg/mL and diluted in a microtitration plate in an appropriate medium, i.e., Cation Adjusted Mueller–Hinton Broth (CaMH, Oxoid, Basingstoke, UK) for staphylococci and Brain Heart Infusion Broth (BHI, Oxoid) for enterococci to reach the final concentration of 256–0.125 µg/mL. Microtitre plates were inoculated with test microorganisms so that the final concentration of bacterial cells was 10^5^. Ampicillin (Sigma) was used as reference drug. A drug-free control and a sterility control were included. The plates were incubated for 24 h at 37 °C for staphylococci and enterococci. After static incubation in the darkness in an aerobic atmosphere, the MIC was visually evaluated as the lowest concentration of the tested compound, which completely inhibited the growth of the microorganism. The experiments were repeated three times. The results are summarized in Table 2.

### 3.5. Determination of Minimum Bactericidal Concentrations

For the above-mentioned strains/isolates, the agar aliquot subculture method [67,68] was used as a test for bactericidal agents. After the MIC value determination, the inoculum was transferred to CaMH (Oxoid) for staphylococci and BHI (Oxoid) for enterococci medium using a multipoint inoculator. The plates were incubated in a thermostat at 37 °C for 24 h. The lowest concentration of test compound at which ≤5 colonies were obtained was then evaluated as MBC, corresponding to a 99.9% decrease in CFU relative to the original inoculum.

### 3.6. MTT Assay

Compounds were prepared as previously stated and diluted in CaMH broth for *S. aureus* to achieve the desired final concentrations. *S. aureus* bacterial suspension in sterile distilled water at 0.5 McFarland was diluted 1:3. Inocula were added to each well by multi-inoculator. Diluted mycobacteria in broth free from inhibiting compounds were used as the growth control. All compounds were prepared in duplicate. Plates were incubated at 37 °C for 24 h for *S. aureus*. After the incubation period, 10% well volume of MTT (3-(4,5-dimethylthiazol-2-yl)-2,5-diphenyltetrazolium bromide) reagent (Sigma) was mixed into each well and incubated at 37 °C 1 h for *S. aureus*. Then, 100 µL of 17% sodium dodecyl sulfate in 40% dimethylformamide was added to each well. The plates were read at 570 nm. The absorbance readings from the cells grown in the presence of the tested compounds were compared with uninhibited cell growth to determine the relative percent inhibition. The percent inhibition was determined through the MTT assay. The percent viability is calculated through the comparison of a measured value and that of the uninhibited control: % viability = OD_570E_/OD_570P_ × 100, where OD_570E_ is the reading from the compound-exposed cells, while OD_570P_ is the reading from the uninhibited cells (positive control). Cytotoxic potential is determined by the percent viability of <70% [59,60]. The results are summarized in Table 3.

### 3.7. Crystal Violet Uptake

The method of crystal violet uptake [63,64] was used to study membrane alteration. A bacterial suspension was cultivated to logarithmic phase in CaMH and harvested at 4500 rpm for 5 min. The cells were washed twice and resuspended in phosphate-buffered saline (PBS) containing 4× MIC of the tested compounds. Tween 20 (1% solution) was used as the positive control. A growth control without antibiotics was included. The tubes were cultivated at 37 °C for 1 h. After that, the tubes were centrifuged at 4500 rpm for 15 min and washed twice in PBS. The cells were resuspended in PBS containing crystal violet (10 µg/mL). After 15 min, the tubes were incubated at 37 °C and centrifuged (15 min, 4500 rpm), and the absorbance of the supernatant at 595 nm was measured. The experiment was repeated five times, and the results were averaged. The percentage of crystal violet uptake was evaluated according to the following equation:$$\%\;of\;uptake=\frac{OD_{595}\;of\;sample}{OD_{595}\;of\;crystal\;violet\;solution\;}\times 100$$

### 3.8. In Vitro Antimycobacterial Evaluation

The evaluation of in vitro antimycobacterial activity of the compounds was performed against *Mycobacterium tuberculosis* ATCC 25177/H37Ra and *M. smegmatis* ATCC 700084. *M. tuberculosis* was grown in Middlebrook Broth (MB), supplemented with Oleic Acid–Albumin–Dextrose–Catalase (OADC) supplement (Difco, Lawrence, KS, USA). At log phase growth, a culture sample (10 mL) was centrifuged at 15,000 rpm/20 min using a benchtop centrifuge (MPW-65R, MPW Med Instruments, Warszawa, Poland). Following the removal of the supernatant, the pellet was washed in fresh MB and resuspended in fresh, ODAC-supplemented MB (10 mL). The turbidity was adjusted to match McFarland standard No. 1 (3 × 10^8^ CFU) with MB. A further 1:10 dilution of the culture was then performed in MB broth. The antimicrobial susceptibility of *M. tuberculosis* was investigated in a 96-well plate format. In these experiments, sterile deionized water (300 µL) was added to all outer-perimeter wells of the plates to minimize evaporation of the medium in the test wells during incubation. Each evaluated compound (100 µL) was incubated with *M. tuberculosis* (100 µL). Dilutions of each compound were prepared in duplicate. For all synthesized compounds, final concentrations ranged from 256 to 0.125 μg/mL. All compounds were dissolved in DMSO, and subsequent dilutions were made in supplemented MB. The plates were sealed with Parafilm and incubated at 37 °C for 14 days. Following incubation, a 10% addition of alamarBlue (Difco) was mixed into each well, and readings at 570 nm and 600 nm were taken, initially for background subtraction and subsequently after 24 h reincubation. The background subtraction is necessary for strongly colored compounds, where the color may interfere with the interpretation of any color change. For noninterfering compounds, blue color in the well was interpreted as the absence of growth, and a pink color was scored as growth.

For *M. smegmatis,* the broth dilution micro-method in Middlebrook 7H9 medium (Difco) supplemented with ADC Enrichment (Difco) was used. The compounds were dissolved in DMSO (Sigma), and the final concentration of DMSO did not exceed 2.5% of the total solution composition. The final concentrations of the evaluated compounds ranging from 256 to 0.125 μg/mL were obtained by twofold serial dilution of the stock solution in a microtiter plate with sterile medium. Bacterial inocula were prepared by transferring colonies from culture to sterile water. The cell density was adjusted to 0.5 McFarland units using a densitometer (Densi-La-Meter, LIAP, Riga, Latvia). The final inoculum was made by 1:1000 dilution of the suspension with sterile water. Drug-free controls, sterility controls, and controls consisting of medium and DMSO alone were included. The determination of results was performed visually after three days of static incubation in the darkness at 37 °C in an aerobic atmosphere.

Isoniazid (Sigma) was used as the positive control, as it is a clinically used antitubercular drug. The minimum inhibitory concentrations (MICs) were defined as the lowest concentration of the compound at which no visible bacterial growth was observed. The MIC value is routinely and widely used in bacterial assays and is a standard detection limit according to the CLSI [65,66]. The results are summarized in Table 2.

### 3.9. In Vitro Cell Viability Analysis

Human monocytic leukemia THP-1 cells obtained from the European Collection of Cell Cultures (ECACC, Salisbury, UK) were used for in vitro antiproliferative assays. They were routinely cultured in Roswell Park Memorial Institute RPMI 1640 medium supplemented with 10% fetal bovine serum (FBS), 2% l-glutamine, 1% penicillin, and streptomycin (all from Sigma-Aldrich) at 37 °C with 5% CO_2_. The cells were incubated in a complete medium (i.e., containing FBS) with test compounds at 37 °C with 5% CO_2_ for 24 h, similarly as we described previously [69]. The effect of the test compounds dissolved in DMSO on cell viability was determined using a Cell Counting Kit-8 (CCK-8; Sigma) according to the manufacturer’s instructions. The results are shown in Table 2.

## 4. Conclusions

Based on the excellent antibacterial activities of 5-chloro-2-hydroxy- *N*-[(2*S*)-3-methyl-1-oxo-1-{[4-(trifluoromethyl)phenyl]amino}butan-2-yl]benzamide (**3c**), 10 new compounds were designed, and together with 11 previously prepared, all 22 agents were tested for antibacterial activity against staphylococcal, enterococcal, and mycobacterial strains. All compounds were also evaluated for their anticancer activity on the human cancer cell line THP-1, with IC_50_ ranging widely from 1.4 to >10 µM. Intermediates **2a**–**h** and **5a**–**c** with a benzyl-protected salicylic hydroxyl were found to be biologically inactive. The final 11 compounds could be divided into eight diamides and three triamides according to the number of amide groups. All the triamides were antibacterially completely inactive, i.e., the prolongation of the diamides by inserting the second amino acid into the linker did not increase the antibacterial activity. The crystal violet uptake assay showed no damage to the bacterial wall/membrane by the active diamides, indicating that the compounds do not affect membranes. On the other hand, interactions of the compounds with the bacterial enzyme equipment can be predicted, as demonstrated for 5-chloro-*N*-[(2*S*)-3-cyclohexyl-1-oxo-1-{[4-(trifluoromethyl)phenyl]- amino}propan-2-yl]-2-hydroxybenzamide (**3g**), 5-chloro-2-hydroxy-*N*-[(2*S*)-(4-methyl- 1-oxo-1-{[4-(trifluoromethyl)phenyl]amino)pentan-2-yl)benzamide (**3f**), 5-chloro- 2-hydroxy-*N*-[(2*S*)-4-(methylsulfanyl)-1-oxo-1-{[4-(trifluoromethyl)phenyl]amino}butan-2-yl]benzamide (**3e**), and 5-chloro-2-hydroxy-*N*-[(2*S*)-(1-oxo-3-phenyl- 1-{[4-(trifluoro- methyl)phenyl]amino}propan-2-yl)benzamide (**3h**) by the MTT assay. It was confirmed that for the compounds to have antibacterial activity, the connecting fragment between the chlorsalicylic and 4-CF_3_-anilide cores must be substituted with a bulky and/or lipophilic chain such as isopropyl, isobutyl, or a thiabutyl chain (compounds **3c**, **3e**, **3f**). Diamides **3e** and **3f** demonstrated activity against all tested strains/isolates comparable to or better than that of the used standards (ampicillin and isoniazid). Antistaphylococcal activities were in the MIC range of 0.070–8.95 μM, anti-enterococcal activities were in the MIC range of 4.66–35.8 μM, and activities of compounds **3f** and **3e** against *M. tuberculosis* and *M. smegmatis* showed MICs of 18.7 and 35.8 μM, respectively. All the effective compounds demonstrated bactericidal activities. Compounds **3g** and **3h**, i.e., diamides substituted with bulkier and lipophilic fragments (cyclohexylmethyl or benzyl), showed only an antistaphylococcal effect. Cytotoxic activity increases with lipophilicity. It can be stated that the high antibacterial activity is associated with a significant cytotoxic effect against cancer cell lines, which makes the investigated diamides interesting anti-invasive agents with dual (cytotoxic and antibacterial) activity. However, this action should be targeted due to the supposed negative effect on healthy tissue and cells.

## Data Availability

The data presented in this study are available on request from the corresponding author.

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
