# Peer review of "Study of Biological Activities and ADMET-Related Properties of Salicylanilide-Based Peptidomimetics"

_ijms, 2022, doi:10.3390/ijms231911648_

Round 1
Reviewer 1 Report
This manuscript details the synthesis and biological evaluation of some salicylanilide containing diamides and triamides. These compounds have been evaluated for antibacterial, antitumor and antihelminithic properties, as well as some physiochemical properties to determine druglikeness. Overall some of the new structures have notable activity, with compound 3f being especially active. Not much is known about how these molecules exert their effects, but future studies may be able to determine the target mechanism. The compounds are well characterized, but it would be good to include copies of the 1H and 13C NMR spectra in the SI, at least for the most active compounds (probably 3d-3g) so others can easily reproduce the work.
Author Response
Dear Reviewer,
Thank you very much for the revision of our manuscript (ijms-1940637) entitled “Study of Biological Activities and ADMET-Related Properties of Salicylanilide-based Peptidomimetics”. We would like to thank you for your kind comments, which helped us to improve our manuscript. Your helpful suggestions were of great value for the preparation of the revised version. They also helped us to improve the professionality as well as the readability of our manuscript.
Please find below your comments (in italic) followed by our answers.
- The compounds are well characterized, but it would be good to include copies of the 1H and 13C NMR spectra in the SI, at least for the most active compounds (probably 3d-3g) so others can easily reproduce the work.
Answer: We agree with these notes. 1H, 13C and 19F NMR spectra of selected compounds (3c, 3d, 3e, 3f, 3g) were added to revised version of Supporting information.
Thank you for considering our revised manuscript.
Aleš Imramovský

Reviewer 2 Report
In this manuscript, a series of salicylanilide-based derivatives were synthesized and evaluated for their antibacterial and anticancer activity. Diamides 3e-h demonstrated promising antibacterial activity, and the lipophilicity is positively correlated with the cytotoxicity.
Major points to be addressed:
· The authors demonstrated that 11 intermediates and 11 targeted compounds were tested for their biological activities, while only result for 11 targeted compounds was listed in the result section. Data for the derivatives 2 and 5 should be included in the manuscript or supplemental file as well, even they are inactive.
· In the section of conclusion, this series was designed based on the compound 3c, which should be described in the compound design paragraph (last paragraph of introduction), regarding 3c as the lead compound.
· The authors showed the Ro5 parameters for the designed compounds, and 6a-c exceeded the recommended values. Why the authors synthesized them even the compounds did not have a good score. The Ro5 parameters should be considered for the design.
· Table 1 was not well presented. 3a and 3c have one row of data, while others have two rows for each. Which one is MIC? or MBC? To be reader friendly, MIC/MBC ratio and the R1 and R2 substituents are recommended to add.
· In section 2.3, ‘IC50 for K562 and MCF-7’ ranged…, where is the data? To test the anticancer activity, more cancer cell lines should be introduced.
· The authors should check the manuscript carefully to avoid typo and grammatical errors. e.g.
1. In section 2.1, ‘coupled with…to yielded 2a-h’
2. legend for table 2, ‘Enterococcus faecalis ATCC-29213’ should be 29212
3. ‘APM’ in table 2 and 3 should be “AMP”
Author Response
Dear Reviewer,
Thank you very much for the revision of our manuscript (ijms-1940637) entitled “Study of Biological Activities and ADMET-Related Properties of Salicylanilide-based Peptidomimetics”. We would like to thank you for your kind comments, which helped us to improve our manuscript. Your helpful suggestions were of great value for the preparation of the revised version. They also helped us to improve the professionality as well as the readability of our manuscript.
Please find below your comments (in italic) followed by our answers.
In this manuscript, a series of salicylanilide-based derivatives were synthesized and evaluated for their antibacterial and anticancer activity. Diamides 3e-h demonstrated promising antibacterial activity, and the lipophilicity is positively correlated with the cytotoxicity.
Major points to be addressed:
- The authors demonstrated that 11 intermediates and 11 targeted compounds were tested for their biological activities, while only result for 11 targeted compounds was listed in the result section. Data for the derivatives 2 and 5 should be included in the manuscript or supplemental file as well, even they are inactive.
Answer: We agree with these notes. The MIC and IC50 values of compounds 2 and 5 have been added to Table S2 in Supplementary Materials.
- In the section of conclusion, this series was designed based on the compound 3c, which should be described in the compound design paragraph (last paragraph of introduction), regarding 3c as the lead compound.
Answer: The last paragraph of the Introduction has been modified and the term "lead compound" has been introduced for compound 3c.
- The authors showed the Ro5 parameters for the designed compounds, and 6a-c exceeded the recommended values. Why the authors synthesized them even the compounds did not have a good score. The Ro5 parameters should be considered for the design.
Answer: Compounds 6a–c, “triamides”, were prepared as interesting peptide-mimicking structures and primarily as anticancer drugs. It is true that their lipophilicity is higher than the recommended (Ro5) log P <5; however, their peptidomimetic structure shows favorable ADME properties. Due to the very good antimicrobial efficiency of some diamides, it was decided to test also selected triamides (derived from the parent most active diamides) for their antimicrobial properties. Unfortunately, the prolongation of the linker has been found to result in loss of activity. On the other hand, this is a valuable insight, and these compounds will no longer be strictly designed as anti-infective compounds.
Due to log P value >5 for some diamides, it should be noted that this value is close to log P = 5 (depending on the type of software with which log P is calculated) and is significantly exceeded only for compound 3g (R1 = cyclohexylmethyl).
Last but not least, it should be noted that, as stated in the manuscript, holding and "blindly" adhering to Ro5 is not a 100% guarantee that the designed and investigated agent will become a drug.
- Table 1 was not well presented. 3a and 3c have one row of data, while others have two rows for each. Which one is MIC? or MBC? To be reader friendly, MIC/MBC ratio and the R1 and R2 substituents are recommended to add.
Answer: Most probably. Table 2 is meant in this comment. We agree with this note. Table 2 has been modified. We wish that now is more “reader friendly”.
- In section 2.3, ‘IC50 for K562 and MCF-7’ ranged…, where is the data? To test the anticancer activity, more cancer cell lines should be introduced.
Answer: Tests on cancer cell lines and reported IC50 values are not the content of this manuscript, but have been published previously and only the range of IC50 values is cited in this manuscript to illustrate the similarity with the current results in THP-1 cells. The authors apologize for any confusion. The sentence has been modified to make it clear where the mentioned results can be found (including appropriate literature).
- The authors should check the manuscript carefully to avoid typo and grammatical errors. e.g.
- In section 2.1, ‘coupled with…to yielded 2a-h’
- legend for table 2, ‘Enterococcus faecalis ATCC-29213’ should be 29212
- ‘APM’ in table 2 and 3 should be “AMP”
Answer: The authors thank the reviewer for careful reading. The manuscript has been carefully revised and all typographical errors have been removed.
Thank you for considering our revised manuscript.
Aleš Imramovský

Round 2
Reviewer 2 Report
Comments have been addressed